# Effects of Supervised Strength Training on Physical Fitness in Children and Adolescents: A Systematic Review and Meta-Analysis

**DOI:** 10.3390/jfmk10020162

**Published:** 2025-05-06

**Authors:** José M. Moreno-Torres, Juan Alfonso García-Roca, Oriol Abellan-Aynes, Alvaro Diaz-Aroca

**Affiliations:** 1Facultad de Deporte, UCAM Universidad Católica de Murcia, 30107 Murcia, Spain; jmmoreno6@alu.ucam.edu (J.M.M.-T.); adiaz2@ucam.edu (A.D.-A.); 2Centro de Estudios Olimpiicos, Universidad Católica de Murcia, 30107 Murcia, Spain; 3Department of Physical Education and Sports, University of Leon, 24071 León, Spain; 4EDUFISOC Educación Física, Deporte y Sociedad, Universidad Católica de Murcia, 30107 Murcia, Spain

**Keywords:** physical conditioning, strength training, educational programming, children, adolescents

## Abstract

**Objective:** Strength training has gained recognition as an effective method to enhance physical fitness in children and adolescents. Its benefits include improvements in muscular strength, aerobic capacity (VO_2_max), and motor performance. However, the diversity in training protocols and participant characteristics across studies necessitates a comprehensive synthesis of the evidence. The aim of this paper was to analyse the influence of a strength training programme in young people aged 6 to 16 years on different aspects of physical fitness. **Methods:** A search was carried out in the EBSCO, Web of Sciences, and Scopus databases. A total of 634 articles were reviewed, and 22 were finally included in the meta-analysis of articles published between 2013 and 2023 in English or Spanish. Twenty-two studies met the inclusion criteria and were assessed using the AXIS and PEDro tools. Standardised mean differences (SMD) and 95% confidence intervals (CI) were calculated. The variables that were most frequently repeated as criteria for evaluating the effectiveness of strength training were the following: (1) strength of the lower/upper body muscles; (2) VO_2_max; (3) sprint performance. **Results:** Strength training interventions produced statistically significant improvements in all analysed variables. Most effective programmes lasted between 6 and 12 weeks, with 2–3 sessions per week. VO_2_max showed the greatest improvement, followed by upper and lower limb strength, and sprint performance. Heterogeneity ranged from low to moderate. **Conclusions:** Supervised strength training programmes can significantly enhance physical fitness in school-aged children and adolescents. While the included studies varied in design and duration, measurable improvements were commonly observed in interventions lasting at least 6–8 weeks. Future research should explore age- and maturity-related responses through subgroup analyses. Limitations include the exclusion of studies published after 2023 and the wide age range of participants without biological stratification.

## 1. Introduction

Physical and psychological development during childhood and adolescence is a complex and multifaceted process that significantly influences health and well-being throughout life [1]. In this context, strength training has emerged as a valuable tool to promote physical health, athletic performance, and motor development in school-aged populations [2,3]. Traditionally, strength training in children and adolescents was met with scepticism and caution due to concerns about possible adverse effects on bone growth and development. However, recent research has challenged these perceptions and provided robust evidence to support the benefits of including strength training programmes at an early age [4,5].

Strength training consists of exercises for improving muscular strength and endurance through the application of external resistance or self-load [6]. In addition to strengthening muscles, it has other physical benefits such as improved bone mineral density and body composition; improved coordination; and improved emotional and social aspects, such as self-esteem and confidence [7,8]. Therefore, in the last decade, different systematic reviews and meta-analyses have analysed how strength interventions in schoolchildren of different age ranges influence various aspects of physical fitness [3,4,9,10]. In general, it has been documented that strength training, when implemented correctly, is safe and effective for children and adolescents, providing benefits that transcend the physical realm and extend to emotional and social development [8].

In terms of public health, the incorporation of strength training into the school physical education curriculum can play a crucial role in the prevention and control of various health conditions prevalent among young people [11]. Childhood obesity, metabolic syndrome, and other sedentary-related diseases represent significant challenges for health systems worldwide. Strength training can be a strategic intervention to combat these problems by promoting an active and healthy lifestyle from an early age [7,12]. In this regard, there is growing evidence to suggest that school strength-training programmes are not limited to improving physical fitness but can also improve students’ academic performance and mental health [13,14]. Regular resistance exercise has been associated with improved concentration, reduced levels of anxiety and depression, and an increase in self-esteem and perceived personal competence [13,14]. These benefits may translate into a more positive school environment and improved academic outcomes, although these dimensions were not the focus of the present analysis [15].

It is essential, however, that strength training programmes are designed and supervised by trained professionals who understand the specific needs and limitations of children and adolescents [16]. Proper progression of loads, correct exercise technique, and education about the importance of physical activity are crucial to ensure that young people gain maximum benefit from these programmes without risk of injury [8]. Promoting a culture of physical activity from childhood will not only improve the health and performance of young people today, but also lay the foundation for a healthy and active adult life [8,16].

Therefore, the aim of this study was to conduct a systematic review and meta-analysis to examine the effects of supervised strength training programmes on physical fitness variables—specifically VO_2_max, muscular strength, and sprint performance—in healthy children and adolescents aged 6 to 16 years. The analysis included studies conducted both within school environments and in extracurricular or sports-based contexts, with the goal of providing evidence-based insights that can inform educational practice and contribute to the development of effective physical training strategies in youth populations. Although previous position statements have established general guidelines for youth strength training [5,8], the present review adds value by offering an updated quantitative synthesis of its effects on physical fitness outcomes over the last decade, based on supervised interventions in healthy children and adolescents.

## 2. Materials and Methods

The protocol for this systematic review and meta-analysis was registered in PROSPERO (International Prospective Register of Systematic Reviews) under registration number CRD420251029311. The review was conducted following the general principles of the PRISMA 2020 guidelines and adhered to established criteria for evidence-based synthesis.

### 2.1. Search Strategy

For the preparation of the document, two reviewers (J.M.M.T and J.A.G.R) independently searched articles published in Spanish and English, supplemented by a manual search, and retrospectively included references if necessary. The search for articles was carried out inEBSCO, WoS, and SCOPUS databases, with the following search strategy (‘force’ OR ‘Strength’ OR ‘energy’ OR ‘resistance’) AND (‘train*’ OR ‘prepare’ OR ‘preparation’ OR ‘exercise*’ OR ‘training’ OR ‘practice’ OR ‘physical activity’ OR ‘physical exercise’ OR ‘athletic’) AND (‘children’ OR ‘Young’ OR ‘adolescent’) and obtained a total of 634 initial articles. After eliminating duplicate articles (66), excluding language (1), titles considered irrelevant for their introduction (412), summaries (105), and full article reading (28), we obtained a final 22 optimal articles for the systematic review and meta-analysis (Figure 1). In order to obtain these articles, key words such as strength training and schoolchildren were used, taking into account their synonyms. Likewise, limits were introduced for the final collection of articles, including schoolchildren aged 6 to 16 years, strength training within and/or outside the school environment, healthy children without diseases and/or pathologies, as well as articles published between 2013 and 2023. On the other hand, articles were excluded if they included children under 6 or over 16 years of age, introduced supplements or drugs before, during, or after training, did not assess strength training, and did not carry out the training under the supervision of a physical activity professional.

### 2.2. Quality Assessment Data Extraction

The cross-sectional study assessment tool (AXIS tool) [17], accessed online at https://bmjopen.bmj.com/content/6/12/e011458 on 15 October 2024, was used to assess the quality of all included studies, regardless of their design. Each item was evaluated as present (‘Yes’) or absent (‘No’), and the percentage of positive items was calculated. Based on established classifications [18], studies were categorised as follows: 0–50% = high risk of bias; 51–79% = medium risk; and 80–100% = low risk. All 22 studies included in the review were classified as medium risk of bias. The full results of this evaluation are shown in Table 1. In addition, the Physiotherapy Evidence Database (PEDro) scale [19] was applied only to those studies with a randomised controlled trial (RCT) design, as this tool is specifically developed for assessing RCTs. The mean PEDro score obtained for these studies was 7.81, which is considered indicative of high methodological quality. Two reviewers (J.M.M.T. and J.A.G.R.) independently conducted the quality assessments. In case of disagreement, a third reviewer (A.D.A.) was consulted to reach a final decision.

### 2.3. Data Extraction

The data extracted from each of the studies (Table 2) included the study design, number of participants, age of participants, inclusion criteria used in each study, characteristics of the type of intervention, duration, frequency, variables analysed in each study, and the instruments used.

### 2.4. Statistical Analysis

The meta-analysis was performed based on the work of Neyeloff [42]. To perform the meta-analysis, the mean, standard deviation, and sample of the pre–post test changes in each of the variables included in both the experimental and control groups were taken into account. These values allowed us to calculate the standardised mean difference (mean difference), together with its 95% confidence interval (CI), which enabled us to evaluate the magnitude of the effect in each study. Heterogeneity was calculated using Chi^2^, and the inconsistency index (I^2^), derived from Cochran’s Q statistic: negligible heterogeneity, 0–40%; moderate heterogeneity, 30–60%; substantial heterogeneity, 50–90%; and considerable heterogeneity, 75–100%.

In cases where a study reported results for more than one independent experimental group, each group was treated as a separate data entry in the meta-analysis, provided that the interventions and samples were clearly distinct and statistically independent. This was the case for Alves [21] and Marta [32], which included multiple experimental arms with different training modalities and no overlap between groups. This approach allowed for a more nuanced analysis of intervention effects without introducing bias through data duplication.

## 3. Results

### 3.1. Results of Strength Training on Maximal Oxygen Volume

Regarding VO_2_max (Figure 2), it can be seen how each row presents the data for the experimental and control groups, specifying the mean, standard deviation, and the number of participants in each. In the first row from Alves et al. [21], the experimental group has a mean of 1.7 and a standard deviation of 5.26 with 45 participants, while the control group has a mean of 0.2, a standard deviation of 5.38, and 44 participants. The mean difference for this study is 1.50, with a 95% confidence interval (CI) in the range [−0.71, 3.71], indicating that, although there is an improvement in VO_2_max, the interval includes negative values, suggesting that the difference may not be significant in this particular case. However, the overall summary of all studies shows a mean difference of 2.14, with a confidence interval of [1.26, 3.02], indicating a significant mean improvement in VO_2_max in schoolchildren undergoing the strength training programme. In terms of the weight of the studies in the overall analysis, it is observed that the studies by Marta [32,33] have the highest weight, with values ranging from 18.9% to 24.7%, while the two entries by Alves et al. [21] have weights of 15.8% and 17.3%. The heterogeneity analysis in this case, shows values that indicate a very high homogeneity, with Tau^2^ = 0.00, Chi^2^ = 1.45 with four degrees of freedom (*p* = 0.84), and an I^2^ = 0%, which means that there is no significant heterogeneity between the studies, i.e., the results are consistent and do not vary significantly between them. Finally, the test for the overall effect shows a Z value = 4.76 with *p* < 0.00001, indicating a highly significant improvement in VO_2_max of the experimental group over the control. In the forest plot, the confidence intervals for each study are represented by green lines, and the mean difference is represented by green squares whose size reflects the weight of the study in the analysis. The black diamond at the bottom of the plot represents the overall estimate, which is clearly to the right of the 0 line (marking the absence of effect), confirming a significant positive effect of strength training on VO_2_max.

### 3.2. Results of Strength Training in Sprinting

Regarding sprinting (Figure 3), it can be seen how for each study, the mean, standard deviation (SD), and the number of participants in the experimental and control groups are presented, which allows us to evaluate the differences in sprint performance between the groups that performed strength training and those that did not. For example, the study by Alagöz [20] shows that the experimental group had a mean of −0.59 and SD of 0.7 with 15 participants, while the control group had a mean of −0.57 and SD of 0.64 with the same number of participants, resulting in a standardised mean difference of −0.03 with a 95% confidence interval in the range [−0.74, 0.69]. This interval includes the value of 0, indicating that the effect is not statistically significant for this particular study. Similarly, other studies present their respective standardised mean differences and associated confidence intervals, such as the study by Hammami [29], which shows a standardised mean difference of −0.89 with a confidence interval of [−1.42, −0.04], suggesting a significant effect in favour of the experimental group, as the interval does not include the value of 0, indicating that strength training significantly improved sprinting in that study. Overall, the total standardised mean difference is −0.32 with a confidence interval of [−0.60, −0.05], indicating a generalised improvement in sprint performance after strength training, being a statistically significant result. In terms of heterogeneity, we report a Tau^2^ of 0.15, a Chi^2^ of 33.09 with 12 degrees of freedom (*p* = 0.0009), and an I^2^ of 64%, indicating moderate-to-high heterogeneity, i.e., the studies show considerable differences between them. This level of heterogeneity suggests that not all studies are completely consistent, possibly due to differences in the populations studied or in the training methods applied. However, despite this heterogeneity, the test for the overall effect shows a Z-value = 2.32 with *p* = 0.02, which confirms that the effect of strength training on sprinting is statistically significant across the set of studies. The forest plot visualises these standardised mean differences for each study, where the green lines represent confidence intervals and the green squares represent the magnitude of the effect for each study, the size of which reflects the weight in the meta-analysis. The black diamond at the bottom of the graph represents the overall effect, which is clearly to the left of the 0 line, confirming a significant effect of strength training in favour of the experimental group in terms of improved sprint performance.

### 3.3. Results of Strength Training in Upper Limb Strength Improvement

Regarding upper limb strength improvement variable (Figure 4), each article shows the mean, standard deviations (SD) and the total number of participants in both the experimental group (who undertook strength training) and the control group (who did not). These values allow the calculation of the standardised mean difference (mean difference), together with its 95% confidence interval (CI), which allows assessment of the magnitude of the effect in each study. For example, in the study by Alagöz [20], the experimental group has a mean of 2 with a SD of 4.62 and 15 participants, while the control group has a mean of 0.46 with a SD of 2.9, resulting in a standardised mean difference of 0.39 with a CI of [−0.33, 1.11], suggesting a positive but not significant effect, as the CI includes the value 0. A more notable study by Milenkovic [36] shows a standardised mean difference of 0.59 with a CI of [0.07, 1.11], indicating a significant improvement in upper body strength for the experimental group. In the forest plot, each green line represents the CI of a study, with the green squares indicating the magnitude of the individual effect, and the size of the square reflecting the weight of that study in the overall meta-analysis. The black rhombus at the end of the graph represents the overall standardised mean difference, which is 0.26 with a CI of [0.15, 0.36], indicating that strength training has a positive and statistically significant effect on the improvement of upper body strength in schoolchildren. Regarding heterogeneity, the results show a Tau^2^ of 0.01, a Chi^2^ of 40.83 with 36 degrees of freedom (*p* = 0.27), and an I^2^ of 12%, indicating low heterogeneity among the studies, i.e., the results are quite consistent among the included studies. Finally, the overall effect test yields a Z value of 4.91 with a *p* < 0.00001, confirming that the overall effect of strength training on upper body strength improvement in schoolchildren is highly significant.

### 3.4. Results of Strength Training on Lower Limb Strength Improvement

As for the lower limb variable (Figure 5), the mean, standard deviations (SD), and the total number of participants in both the experimental group (which underwent strength training) and the control group (which did not) are presented in detail. These values allow us to calculate the standardised mean difference (mean difference), together with its 95% confidence interval (CI), thus assessing the magnitude of the effect in each study. For example, in the study by Alagöz [20], the experimental group shows a mean of 0.03 with a SD of 5.94 and 15 participants, while the control group has a mean of 0.01 with a SD of 6.03 and 15 participants, resulting in an SMD of 0.00 with a CI of [−0.71, 0.72], suggesting no significant effect, since the CI includes the value 0. Another study, the one by Alves et al. [21], presents a mean of 0 with a SD of 0.14 and 39 participants in the experimental group, and a mean of 0 with a SD of 0.14 in the control group, resulting in an SMD of 0.00 with a CI of [−0.42, 0.42], again indicating a non-significant effect. In contrast, the study by Mesfar [35] shows an SMD of 4.54 with a CI of [3.14, 5.94], indicating a significant improvement in lower body strength in the experimental group. In the forest plot, each green line represents the CI of a study, the green squares indicate the magnitude of the individual effect, and the size of the square reflects the weight that the study has in the overall meta-analysis. The black diamond at the bottom of the graph represents the overall SMD, which is 0.18 with a CI of [0.03, 0.33], indicating that strength training has a positive and statistically significant effect on lower body strength improvement. Regarding heterogeneity, the results show a Tau^2^ of 0.15, a Chi^2^ of 119.47 with 47 degrees of freedom (*p* < 0.00001), and an I^2^ of 61%, indicating moderate heterogeneity between studies. Finally, the overall effect test yields a Z value of 2.37 with a *p* = 0.02, confirming that the overall effect of strength training on lower body strength improvement is statistically significant.

## 4. Discussion

Our systematic review and meta-analysis aimed to examine the effects of strength training on different aspects of physical fitness in children and adolescents. Our meta-analysis identified 22 studies and suggested that strength training can produce small but significant improvements in VO_2_max, upper and lower body strength, and sprinting.

The significant increase in VO_2_max observed in this meta-analysis (mean difference of 2.14, CI of [1.26, 3.02]) is a key finding that underscores the effectiveness of strength training in improving aerobic capacity in schoolchildren. This result is particularly relevant in the context of public health, as higher VO_2_max is associated with better cardiovascular health and greater capacity to perform intense physical activities [34]. The consistency of these results across studies (I^2^ = 0%) suggests that strength training programmes, regardless of their methodological variations, can be an effective tool for improving aerobic capacity in children and adolescents.

Likewise, the results of the meta-analysis also indicate significant improvements in upper (overall standardised mean difference of 0.26, CI [0.15, 0.36]) and lower (overall standardised mean difference of 0.18, CI [0.03, 0.33]) limb strength. The low heterogeneity in the studies on upper limb strength (I^2^ = 12%) suggests a high consistency in the benefits of strength training in this area. On the other hand, the moderate heterogeneity in the studies on lower limb strength (I^2^ = 61%) could be attributed to differences in the intensity and type of exercises employed. Improvement in muscular strength is crucial not only for sports performance, but also for bone health and injury prevention [8]. Increased muscle strength can contribute to better posture, greater joint stability and a reduced risk of injury in both sporting activities and daily life.

On the other hand, the analysis of sprint performance showed an overall standardised mean difference of −0.32 (CI of [−0.60, −0.05]), indicating a significant improvement after the implementation of strength training. However, the moderate heterogeneity (I^2^ = 64%) suggests that the variability in the results may be influenced by factors such as the age of the participants, the duration of the training program, and differences in the specific protocols used. Despite this variability, the overall improvement in sprint performance is consistent with previous studies that have documented improvements in speed and agility as a result of strength training [29]. These training programmes, which typically lasted between 6 and 12 weeks, can be considered mid-term interventions. The observed improvements in VO_2_max, muscular strength, and sprint performance are consistent with the expected physiological adaptations resulting from such medium-duration training cycles in youth populations.

In addition to the statistical significance of the findings, it is important to consider the training load parameters and underlying physiological mechanisms that may explain the improvements observed. Among the studies included in this review, the most common training configurations involved a frequency of 2–3 sessions per week, with durations ranging from 45 to 60 min, moderate-to-high volume (2–4 sets of 6–15 repetitions), and rest intervals of 30–90 s. These parameters reflect moderate training loads, which appear sufficient to elicit meaningful adaptations in children and adolescents while minimising the risk of overtraining.

From a physiological perspective, age and biological maturation play a central role in the nature and extent of training adaptations. In prepubescent children, improvements in strength and performance are primarily attributed to neuromuscular adaptations, such as increased motor unit recruitment, enhanced intermuscular coordination, and reduced coactivation of antagonist muscles [5,43]. These adaptations do not rely on increases in muscle cross-sectional area, as hormonal levels are not yet sufficient to support hypertrophic responses.

During and after puberty, the rise in anabolic hormones, such as testosterone, growth hormone, and IGF-1, facilitates greater muscular hypertrophy and power output [8,44]. Consequently, adolescents may exhibit more pronounced physical improvements in response to strength training than younger children. These developmental differences highlight the need for training programmes to be adapted according to the individual’s stage of biological maturity rather than strictly by chronological age [5]. In this context, the observed improvements in speed and agility gain further relevance, as they may contribute to better athletic development and injury prevention, as improved speed and agility may contribute to better performance in various sports and physical activities.

Overall, the present study has evidenced the efficacy of strength training in improving the physical performance and general condition of schoolchildren, allying with the findings of other studies in the area [5,45]. However, it is crucial to perform a meticulous analysis of the different types of strength training and their specific effects on schoolchildren, considering their various maturation levels. There are divergences in the literature on the safety and efficacy of strength training in children and adolescents. For example, Faigenbaum and Myer [5] argue that strength training is safe and beneficial for children when performed with proper technique and adequate supervision. In contrast, Malina [43] expresses concerns about the potential for injury and the need for strict supervision to minimize risks. Lloyd [45] argue that strength training may be particularly beneficial during puberty due to hormonal surges that facilitate muscle development. However, Faigenbaum and Myer [5] stress the importance of tailoring training programmes to biological and not just chronological age, emphasizing that an individualised approach is crucial to maximize benefits and minimize risks. Other studies, such as that of Granacher [44] suggest that the effects of strength in-training on improving physical performance may be less pronounced in younger children compared to adolescents due to differences in neuromuscular maturation.

Furthermore, based on the analysis of the included studies and in line with the foundational literature on youth strength training [8,46,47], we identified substantial similarities between extracurricular and school-based training programmes in children and adolescents aged 6 to 16. These similarities include session duration and frequency, the use of playful or pedagogically guided activities, and the emphasis on supervised, age-appropriate training. This conceptual and methodological convergence supports the decision to consider both contexts jointly in this review and reinforces the transferability of the results to educational settings.

The findings of the present work have important practical implications for coaches, physical education teachers, and health professionals working with schoolchildren. The implementation of strength training programmes, tailored to children’s individual maturation levels and needs, can contribute significantly to their physical development and overall well-being [5,48]. It is crucial to ensure that these programmes are properly supervised, and that emphasis is placed on correct technique to minimize the risk of injury. Future research should continue to explore the best practices for the implementation of strength training in school settings, contributing to the development of evidence-based guidelines and the overall improvement of the schoolchild in particular and the educational curriculum at general levels.

## 5. Conclusions

Based on the data collected from the scientific literature, following which the present meta-analysis was conducted, it has been shown that the implementation of strength training programmes in schoolchildren produces significant improvements in several key areas of physical fitness. In particular, notable increases were observed in maximal volume of oxygen (VO_2_max), upper and lower limb strength, and sprint performance.

The data collected and analysed demonstrate that strength training is an effective intervention for improving VO_2_max, suggesting improved aerobic capacity in schoolchildren. The improvement in upper and lower limb strength was also significant, indicating that this type of training contributes to the development of overall muscle strength in this population. Furthermore, the results in sprint performance show that strength training not only benefits aerobic capacity and muscle strength, but also improves speed and acceleration capacity in schoolchildren.

These findings highlight the importance of including a minimum of 8-week strength training programmes in the educational curriculum to promote the health and overall physical development of students. Under the premise of adequate supervision, practical guidelines that prioritize self-loading exercises or accessible and versatile equipment, such as elastic bands or dumbbells, strength training programmes can be adapted to different skill levels and specific needs.

## Figures and Tables

**Figure 1 jfmk-10-00162-f001:**
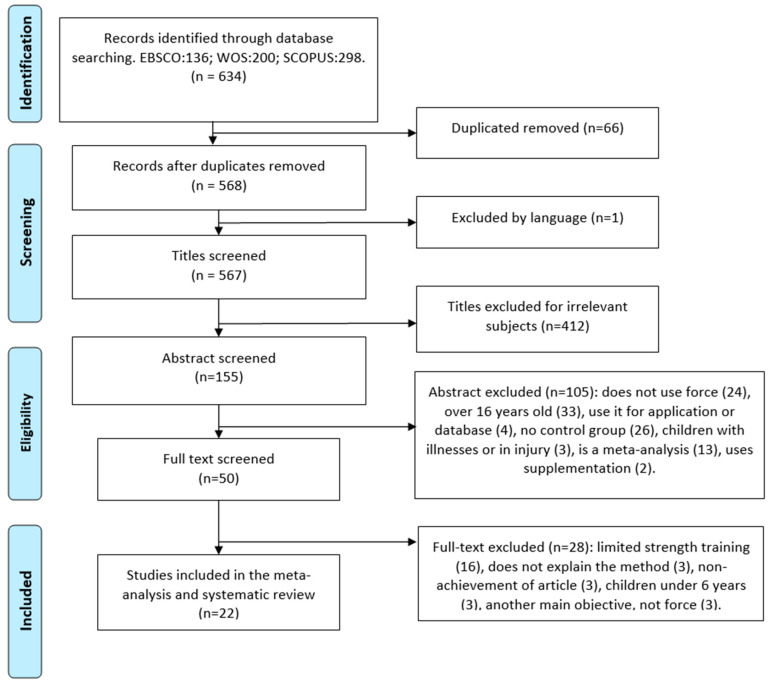
PRISMA flow chart for systematic reviews with database searches.

**Figure 2 jfmk-10-00162-f002:**
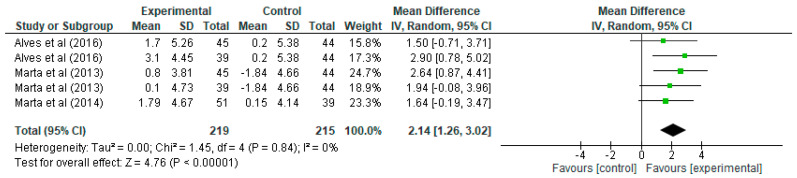
Influence of strength training in schoolchildren on VO_2_max. Note: Alves [21] and Marta [32] appear more than once in the figure as they report data from distinct intervention groups analysed separately [33].

**Figure 3 jfmk-10-00162-f003:**
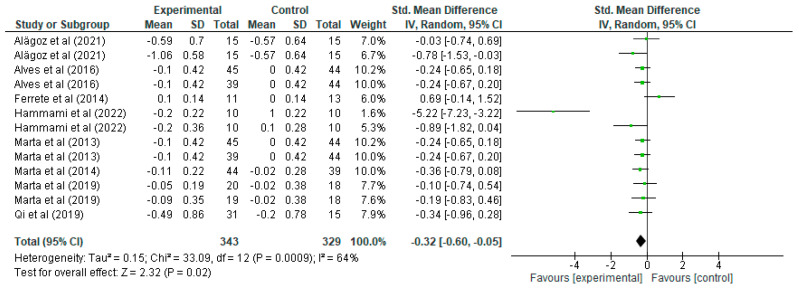
Influence of strength training in schoolchildren on sprinting [20,21,27,29,31,32,33,39].

**Figure 4 jfmk-10-00162-f004:**
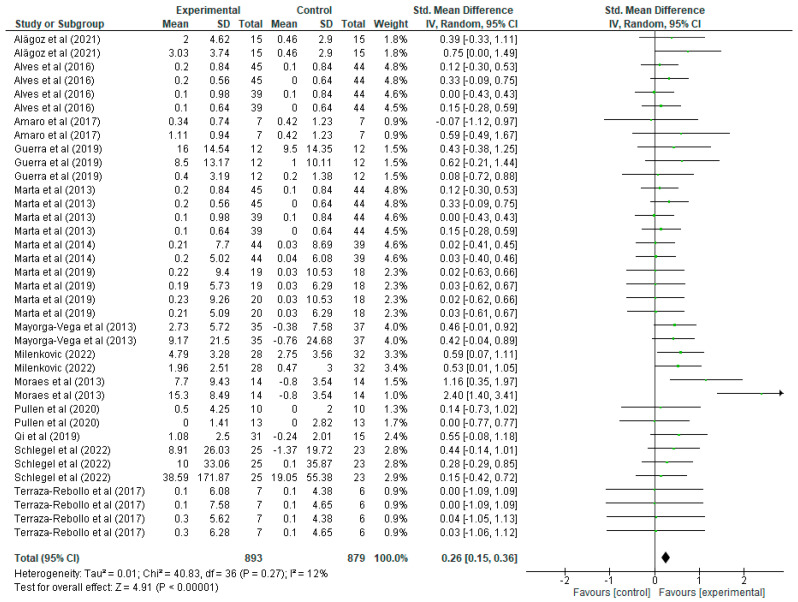
Influence of strength training in schoolchildren on upper limb strength improvement. Note: Alves [21] and Marta [32] appear more than once in the figure as they report data from distinct intervention groups analysed separately [20,22,28,31,33,34,36,37,38,39,40,41].

**Figure 5 jfmk-10-00162-f005:**
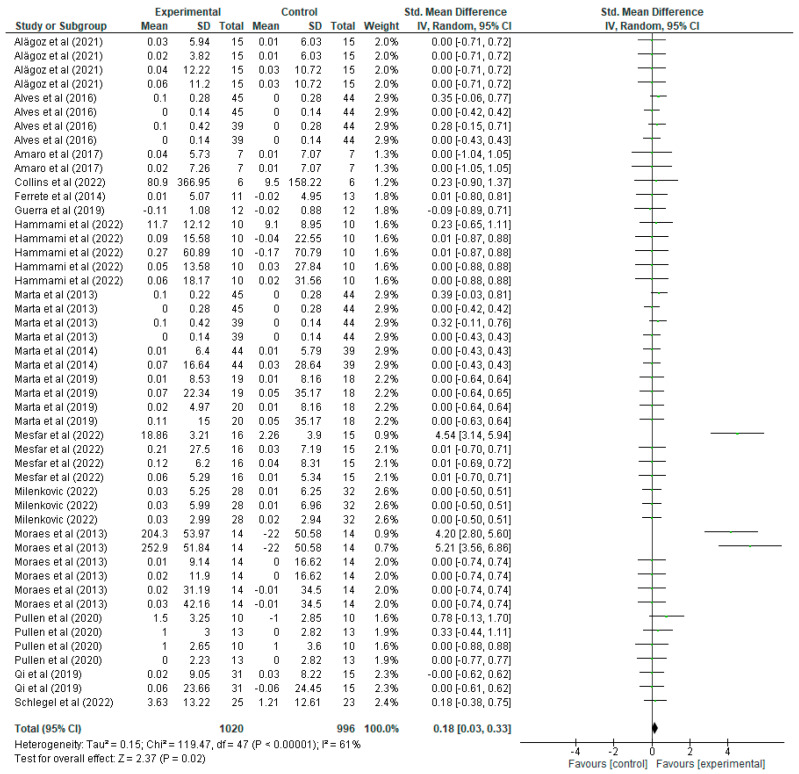
Influence of strength training in schoolchildren in lower limb strength improvement. Note: Alves [21] and Marta [32] appear more than once in the figure as they report data from distinct intervention groups analysed separately [20,22,25,27,28,29,31,33,35,36,37,38,39,40].

**Table 1 jfmk-10-00162-t001:** Quality appraisal of the included cross-sectional studies according to the AXIS tool.

Studies	1	2	3	4	5	6	7	8	9	10	11	12	13	14	15	16	17	18	19	20	YES	No	%YES	Risk of Biais
[20]	YES	YES	NO	YES	YES	YES	NO	YES	YES	YES	YES	YES	NO	NA	YES	YES	YES	YES	NO	YES	15	4	79%	MEDIUM
[21]	YES	YES	NO	YES	YES	YES	NO	YES	YES	YES	YES	YES	NO	NA	YES	YES	YES	YES	NO	YES	15	4	79%	MEDIUM
[22]	YES	YES	NO	YES	YES	YES	NO	YES	YES	YES	YES	YES	NO	NA	YES	YES	YES	YES	NO	YES	15	4	79%	MEDIUM
[23]	YES	YES	NO	YES	YES	YES	NO	YES	YES	YES	YES	YES	NO	NA	YES	YES	YES	YES	NO	YES	15	4	79%	MEDIUM
[24]	YES	YES	NO	YES	YES	YES	NO	YES	YES	YES	YES	YES	NO	NA	YES	YES	YES	NO	NO	YES	14	5	74%	MEDIUM
[25]	YES	YES	NO	YES	YES	YES	NO	YES	YES	YES	YES	YES	NO	NA	YES	YES	YES	YES	NO	YES	15	4	79%	MEDIUM
[26]	YES	YES	NO	YES	YES	YES	NO	YES	YES	YES	YES	YES	NO	NA	YES	YES	YES	YES	NO	YES	15	4	79%	MEDIUM
[27]	YES	YES	NO	YES	YES	YES	NO	YES	YES	YES	YES	YES	NO	NA	YES	YES	YES	YES	NO	YES	15	4	79%	MEDIUM
[28]	YES	YES	NO	YES	YES	YES	NO	YES	YES	YES	YES	YES	NO	NA	YES	YES	YES	YES	NO	YES	15	4	79%	MEDIUM
[29]	YES	YES	NO	YES	YES	YES	NO	YES	YES	YES	YES	YES	NO	NA	YES	YES	YES	YES	NO	YES	15	4	79%	MEDIUM
[30]	YES	YES	NO	YES	YES	YES	NO	YES	YES	YES	YES	YES	NO	NA	YES	YES	YES	YES	NO	YES	15	4	79%	MEDIUM
[31]	YES	YES	NO	YES	YES	YES	NO	YES	YES	YES	YES	YES	NO	NA	YES	YES	YES	YES	NO	YES	15	4	79%	MEDIUM
[32]	YES	YES	NO	YES	YES	YES	NO	YES	YES	YES	YES	YES	NO	NA	YES	YES	YES	YES	NO	YES	15	4	79%	MEDIUM
[33]	YES	YES	NO	YES	YES	YES	NO	YES	YES	YES	YES	YES	NO	NA	YES	YES	YES	YES	NO	YES	15	4	79%	MEDIUM
[34]	YES	YES	NO	YES	YES	YES	NO	YES	YES	YES	YES	YES	NO	NA	YES	YES	YES	YES	NO	YES	15	4	79%	MEDIUM
[35]	YES	YES	NO	YES	YES	YES	NO	YES	YES	YES	YES	YES	NO	NA	YES	YES	YES	YES	NO	YES	15	4	79%	MEDIUM
[36]	YES	YES	NO	YES	YES	YES	NO	YES	YES	YES	YES	YES	NO	NA	YES	YES	YES	YES	NO	YES	15	4	79%	MEDIUM
[37]	YES	YES	NO	YES	YES	YES	NO	YES	YES	YES	YES	YES	NO	NA	YES	YES	YES	YES	NO	YES	15	4	79%	MEDIUM
[38]	YES	YES	NO	YES	YES	YES	NO	YES	YES	YES	YES	YES	NO	NA	YES	YES	YES	YES	NO	YES	15	4	79%	MEDIUM
[39]	YES	YES	NO	YES	YES	YES	NO	YES	YES	YES	YES	YES	NO	NA	YES	YES	YES	YES	NO	YES	15	4	79%	MEDIUM
[40]	YES	YES	NO	YES	YES	YES	NO	YES	YES	YES	YES	YES	NO	NA	YES	YES	YES	YES	NO	YES	15	4	79%	MEDIUM
[41]	YES	YES	NO	YES	YES	YES	NO	YES	YES	YES	YES	YES	NO	NA	YES	YES	YES	YES	NO	YES	15	4	79%	MEDIUM

Note: 1 = Were the aims/objectives of the study clear; 2 = Was the study design appropriate for the stated aim(s); 3 = Was the sample size justified; 4 = Was the target/reference population clearly defined? (Is it clear who the research was about?); 5 = Was the sample frame taken from an appropriate population base so that it closely represented the target/reference population under investigation; 6 = Was the selection process likely to select subjects/participants that were representative of the target/reference population under investigation; 7 = Were measures undertaken to address and categorise non-responders; 8 = Were the risk factor and outcome variables measured appropriate to the aims of the study; 9 = Were the risk factor and outcome variables measured correctly using instruments/measurements that had been trialled, piloted or published previously; 10 = Is it clear what was used to determined statistical significance and/or precision estimates? (eg, *p* values, CIs); 11 = Were the methods (including statistical methods) sufficiently described to enable them to be repeated; 12 = Were the basic data adequately described; 13 = Does the response rate raise concerns about non-response bias; 14 = If appropriate, was information about non-responders described; 15 = Were the results internally consistent; 16 = Were the results for the analyses described in the methods, presented; 17 = Were the authors’ discussions and conclusions justified by the results; 18 = Were the limitations of the study discussed; 19 = Were there any funding sources or conflicts of interest that may affect the authors’ interpretation of the results; 20 = Was ethical approval or consent of participants attained.

**Table 2 jfmk-10-00162-t002:** Characteristics of the studies included in the systematic review.

Author	Design, Sample, Age	Inclusion Criteria	Intervention	Time, Freq, Duration, Rep, Sets	Variables	Instruments
[20]	Design: Quasi-experimental Sample: EG = 30; CG = 15 Age: EG = 9.0 ± 0.6; CG = 9.1 ± 0.7	Children who regularly practice swimming, aged between 9 and 11 years, with 2 years of experience.	E = Swimming training program and body strength program using bodyweight exercises, medicine ball, and resistance bands. C = Regular swimming training.	T = 12 weeks F = 3/weeks D = 90 (Strength + swimming) R = 8–12 S = 3	Biceps, chest, waist, hip, and thigh circumference. Vertical jump, flexed arm strength, speed, upper body strength, horizontal jump, flexibility, aerobic endurance, balance, and 50 m freestyle swimming were also measured.	Anthropometric tests with their respective equipment. UBS test for ball throw. FAS test for static arm strength. Sit and reach test for spinal flexibility. Cooper test.
[21]	Design: RCT Sample: EG = 84; CG = 44 Age: EG = 10.5 ± 0.8; CG = 10.6 ± 0.9	Aged 10 to 12 years without chronic paediatric disease or orthopaedic limitations and without regular extracurricular physical activity (i.e., participation in a sport at an academy).	E = Strength and aerobic training before or after strength training. C = Physical education classes only.	T = 8 weeks F = 2/week D = 45–90 min S = 2–4 R = 4–8	Strength, VO_2_max, and anthropometric variables.	Anthropometric instruments, Eurofit test battery, contact mat, and photocells.
[22]	Design: Controlled trial Sample: EG = 14; CG = 7 Age: EG = 12.7 ± 0.8; CG = 12.7 ± 0.8	Two years of experience in swimming.	E = Strength training and swimming. C = Strength training only.	T = 10 weeks. F = 2/week D = 30 min Before swimming S = 3 R = as many as possible in 40 s	Strength, mechanical impulse, vertical jump, and ball throw.	Contact mat, medicine ball, stopwatch.
[23]	Design: RCT Sample: EG-LRM = 17; EG-HRM = 16; CG = 12 Age: EG = 13.7 ± 0.8; CG = 13.7 ± 0.8	The potential participants had to be between 13 and 15 years old, have never participated in a resistance training program, and have no health issues.	E = Leg press, knee extension, barbell bench press, dumbbell fly, lat pulldown, seated row, crunches, and leg raises. C = Without training.	T = 9 weeks F = 2/week. D = 1 h S = 2 R = 4–6 y 12–15.	Upper and lower body muscle strength.	Machinery for the exercises described.
[24]	Design: Controlled trial Sample: EG = 16; CG = 16 Age: EG = 11.5 ± 0.6; CG = 11.9 ± 1.2	Healthy children without diagnosed diseases.	E = Strength program for upper and lower body. C = Without training.	T = 8 weeks F = 3/week D = NR S = 4–8 y 1–2 R = 5 y 20 (First, a high-load and low-repetition group, followed by a high-repetition group)	Unilateral, ipsilateral, and contralateral strength of the lower limbs, power, and endurance (one-repetition maximum (1RM) leg press, knee extensors (KE) and flexors (KF) maximum voluntary isometric contractions (MVIC), countermovement jump (CMJ), muscle endurance test (leg press repetitions at 60% of 1RM)), and unilateral upper body elbow MVIC.	Horizontal press measuring 1RM. Isometric quadriceps and hamstring extension. Dynamometer. Ergojump for CMJ. Hand dynamometer for arm strength.
[25]	Design: Quasi-experimental Sample: EG = 6; CG = 6 Age: EG = 8.9 ± 1.0; CG = 8.9 ± 1.0	Students aged 8 to 10 years, with no current participation in resistance training, no pathology or disability affecting movement, behaviour, or neuropsychological function, and no physical injury.	E = Strength training with deadlift, squat, row, lunge, pull-ups, and hollow hold, along with a warm-up and cool-down. C = Normal physical activity, without resistance training.	T = 10 week F = 2/week D = 45 min S = 2 R = 6–8	Perceived strength, physical self-assessment, global self-esteem, sports competence, physical condition, FMS, height, body mass, BMI, skinfolds, hip, waist, and arm circumference, maximum strength, and relative strength.	Anthropometric instruments, feedback session, and questionnaires (CY-PSPP), Actigraph GT3X+ accelerometer, FMS (Canadian assessment of agility and movement skills), extension machine for performing an isometric mid-thigh pull to measure strength.
[26]	Design: RCT Sample: EG = 11; CG = 11 Age: EG = 15.4 ± 0.9; CG = 14.5 ± 0.8	Healthy subjects without injuries or pathologies.	E = Full-body strength training. C = Normal life.	T = 4 week F = 2/week D = 60 min S = 3–4 R = 10–15	Strength through the execution of the back squat. Peak squat velocity.	Necessary equipment for squat execution and anthropometric materials for muscle mass and height measurement, as well as an accelerometer for peak velocity.
[27]	Design: Controlled trial Sample: EG = 11; CG = 13 Age: EG = 9.32 ± 0.25; CG = 8.26 ± 0.33	Subjects with potential medical problems or a history of ankle, knee, or back pathology within 3 months prior to the start of the study.	E = strength training with ¼ squat, weighted vertical jump, deep squat, push-ups, hurdle jumps, and sprints. C = normal soccer training.	T = 26 week. F = 2/week D = 30 min S = 2–4 R = 4–8	Strength, speed, endurance and flexibility.	Anthropometric instruments, CMJ with ergojump contact platform, photocells for sprint speed, YO-YO test and sit-and-reach test.
[28]	Design: Controlled trial Sample: EG = 12; CG = 12 Age: EG = 15.0 ± 0.84; CG = 15.0 ± 0.84	Students without sports practice.	E = 6-station strength circuit, including squats, push-ups, triceps dips, horizontal jump, plank and “the rower” curl up. C = physical education classes without strength training, they did stretch.	T = 8 weeks F = 2/week. D = 50 min S = 3 R = 30 s of execution of each of the exercises, as many repetitions as possible.	Strength and power in upper and lower body.	Meter and stopwatch.
[29]	Design: RCT Sample: EG = 10; CG = 10 Age: EG = 11.1 ± 0.8; CG = 11.1 ± 0.8	Weightlifting children.	E = eccentric training program focusing on hamstrings with glute and hamstring raises, glute push-ups, good mornings, one-leg Romanian deadlifts, and manual glute raises. C = normal weightlifting training.	T = 6 weeks F = 2/week D = 60–90 min S = 3–5 R = 10–12	Sprint, balance, muscle strength, power and asymmetry between lower limbs.	Sprint with photocells. Horizontal jump power meter. Squat bar for measuring 1RM to measure muscle strength. Asymmetry meter using a single leg jump. Y-balance test, apparatus.
[30]	Design: RCT Sample: EG = 42; CG = 63 Age: EG = 11.63 ± 0.97; CG = 11.64 ± 0.87	Apparently healthy children.	E = high intensity strength program. C = normal physical education class.	T = 8 weeks F = 2/week D = 60 min S = 2 R = It was carried out by execution time, week 1 and 2 (30 s), week 3 and 5 (40 s) and week 6 and 8 (45 s).	Cardiorespiratory variables and certain aspects related to muscle strength were measured. Fat mass.	Fat mass and body mass with bioimpedance. Physical fitness with fitnessgram test. Walking and return tests with cones to assess cardiorespiratory fitness. SLJ tests to assess the power of the lower limbs. And curl up, push up, standing long jump, and vertical jump to assess strength.
[31]	Design: RCT Sample: ER = 19; ES = 20; CG = 18 Age: ER = 10.6 ± 0.5; ES = 10.7 ± 0.5; CG = 10.6 ± 0.5	Children between 10 and 11 years old, physically active, in optimal health and who regularly attend training sessions.	E = strength training with conventional and TRX exercises. C = conventional physical education classes.	T = 8 weeks F = 2/week D = 45 min S = 2–4 R = 8–15	The strength of the lower and upper limbs, as well as motor coordination, was observed.	Contact platform, manual dynamometer, and photocells.
[32]	Design: Controlled trial Sample: ER = 41; ECoN = 45; CG = 39 Age: ER = 10; ECoN = 10; CG = 10	Boys and girls aged 10 to 11.5 years without chronic paediatric disease or orthopaedic limitation and without regular extracurricular PA.	E = strength training and the RYF group was also subjected to a 20 m round trip race after strength training. C = normal physical education.	T = 8 weeks F = 2/week D = 45–60 min S = 2–5 R = 4–8	Body anthropometric measurements were made, explosive strength of the lower and upper body, running speed and VO_2_max were calculated.	CD and audio for the 20 m dash. Contact pad for CJM, cones for the 20 m dash distance, and medicine balls for strength training.
[33]	Design: Controlled trial Sample: ER = 41; ECoN = 45; CG = 39 Age: ER = 10.7 ± 0.4; ECoN = 10.7 ± 0.5; CG = 10.8 ± 0.5	Children aged between 10 and 11.5 years (5th and 6th grade), without chronic paediatric disease or orthopaedic limitation and without regular extracurricular physical activity (e.g., practicing a sport in a club)	ER and CON = After the warm-up period, both GR and GCON groups were subjected to a strength training program consisting of 1 and 3 kg medicine ball throws; box jumps from 0.3 m to 0.5 m; plyometric jumps over a 0.3–0.5 m high hurdle and 30–40 m sprint runs. After finishing strength training for the GR and GCON groups, the GCON group was additionally subjected to a 20 m shuttle run exercise. C = Physical education class.	T = 8 weeks F = 2/week D = 45–60 min S = 2–5 R = 3–8	Strength and VO_2_max measurements were carried out.	Contact platform. Eurofit battery. Photocells for sprinting and viewing the acquired speed. Course navette and formula to calculate Vo2max. Anthropometric tools.
[34]	Design: Controlled trial Sample: EG = 35; CG = 37 Age: EG = 11.10 ± 0.38; CG = 11.10 ± 0.38	Healthy children with no apparent illness.	E = Strength/endurance and cardiovascular circuit program. C = Physical education class.	T = 8 weeks F = 2/week D = 50 min S = 2 R = for time of 15 a 30 s.	Abdominal muscle endurance, upper body endurance, and cardiovascular endurance were measured.	Medicine balls, pull-up bar/wall bars for hanging, and speaker for course navette.
[35]	Design: RCT Sample: EG = 16; CG = 15 Age: EG = 14.4 ± 0.6; CG = 14.5 ± 0.5	Playing volleyball regularly, 3–4 times x week (i.e., ~90 min per session), with one match played during the weekend, for more than 3 years.	E = Resistance training with bench press, pull over, half squat, and forward lunge C = Normal training.	T = 8 weeks F = 2/week D = 50 min S = 3–4 R = 2–4	Dynamic balance was measured. 1RM back squat. CMJ was measured. Lower limb asymmetry.	Y test. Barbell and plates for back squat. Contact platform for CMJ.
[36]	Design: RCT Sample: EG = 28; CG = 32 Age: EG = 13; CG = 13	There were no chronic illnesses or major injuries in the students’ health records.	E = special program for strength development using circuit training within the main part of the class C = regular physical education class.	T = 8 weeks F = 2/week D = 50 min S = 2 R = 30 s	The jump squat and CMJ were measured. Normal squat. Push-ups and sit-ups.	CMJ and sato squat with photocells.
[37]	Design: RCT Sample: EG = 14; CG = 10 Age: EG = 15.4 ± 0.9; CG = 15.4 ± 0.9	Physically active, but had not performed RT; had no functional limitation in performing the prescribed RT program; had no injuries.	E = strength training with bench press on machine, leg press at 45°, front lat pull-down, leg extension, military press, seated leg curl, triceps extension on pulley, sit-ups and arm curl. C = normal life, without ST.	T = 12 weeks F = 3/week D = 60 min S = 3 R = 3–20	1RM strength, power, and flexibility.	Leg press and bench press. Contact platform. Flexibility box.
[38]	Design: RCT Sample: EM = 10; EF = 13; CM = 10; CF = 13 Age: EM = 12.07 ± 0.3; EF = 12.56 ± 0.97; CM = 11.67 ± 0.35; CF = 11.90 ± 0.76	Children from 11 to 14 years old, minimum attendance at 5 sessions.	E = activities based on strength and physical conditioning. C = regular physical education class.	T = 6 weeks F = 3 sessions every 2 weeks. D = 60 min S = 1 R = 5–60 y 30 s according to exercise	Strength, autonomy, global self-esteem and physical self-efficacy.	Measuring meter. Test.
[39]	Design: Controlled trial Sample: EG = 31; CG = 15 Age: EG = 8–12; CG = 8–12	Healthy children.	E = strength-resistance training and plyometrics. Squats, biceps curls, lunges, jumps, sit-ups and running. C = normal physical education classes.	T = 12 weeks F = 2/week D = 60 min S = 2–4 R = 8–30	Anthropometric variables were measured, strength profiles with biceps curl, vertical jump and long jump with one foot and power and speed with a 30 m sprint.	Anthropometric utensils. Metro. Photocells. Chronometer.
[40]	Design: RCT Sample: EG = 25; CG = 23 Age: EG = 10–11; CG = 10–11	Without inclusion criteria, regular school without specialization in strength and/or resistance.	E = Strength training with body weight resistance consisting of push up, hang hold, sit and reach, squat and plank hold. C = Regular physical education classes.	T = 6 weeks F = 2/week D = 45 min S = 1–3 min R = <15	Strength was measured through tests that they had trained for, to see changes from the beginning to the present moment. The tests included push ups, squats, suspended hold, plank, and sit-and-reach.	Pull-up bar, parallel bars, stopwatch, and meter.
[41]	Design: Controlled trial Sample: EG = 14; CG = 6 Age: EG = 14.9 ± 0.7; CG = 15.5 ± 0.9	More than 4 years’ prior experience in tennis training, not practice any other sport, not have participated in any strength training and have not suffered any injuries during the last 6 months.	E = strength training with overload and elastic bands. C = technical-tactical training	T = 8 weeks F = 3/week D = 3 h S = 3 R = 6	It was measured through the improvement in strength as well as how the tennis throwing speed was improved.	radar (Stalker Pro, Newark, CA, USA)

Legend: 1RM = one repetition maximum; BMI = body mass index C = control group; CF = control group, female; CM = control group, male; CON = conventional training group; D = duration; EG = experimental group; EF = experimental group, female; EM = Experimental group, male; ECoN = experimental group with conditioning; ER = endurance training group; F = frequency; GCON = general conditioning group; HRM = high-repetition maximum (low-load); LRM = low-repetition maximum (high-load); NR = not reported; PA = physical activity; R = repetitions; RT = resistance training; RYF = Run-Your-Fit group; S = Sets; ST = strength training; T = total intervention time; VO_2_max = maximum oxygen consumption.

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
