# Peer review of "Effects of Supervised Strength Training on Physical Fitness in Children and Adolescents: A Systematic Review and Meta-Analysis"

_jfmk, 2025, doi:10.3390/jfmk10020162_

Round 1

Reviewer 1 Report

Comments and Suggestions for Authors

The article explores the intriguing topic of the impact of strength training on children and adolescents concerning physical fitness parameters. The title suggests that the article will examine studies related to strength training in children at various developmental stages, regardless of the training context. From the introduction, it appears that the study will assess data from research conducted in a school environment. However, the methodology indicates that studies involving children aged 6-16 years will be analyzed, regardless of whether the research was conducted in a school setting or not.

I believe the aim of the article is unclear. In any case, it would be particularly interesting to analyze studies conducted in the school environment, providing physical education teachers the opportunity to adapt their lesson plans accordingly. Furthermore, the discussion was expected to examine the load parameters and draw conclusions regarding the frequency of training, intensity, volume, and rest intervals. Nevertheless, the discussion focused on whether the results were significant.

Introduction

The introduction focuses on the effects of strength training within the school environment on physical fitness parameters. However, it does not align with the title and methodology of the research.

The authors must determine whether they wish to analyze studies conducted in school settings or examine children in general, including those who are or are not involved in organized sports forms.

Methodology

Major Issues:

  • When performing the advanced search in Scopus, I am unable to reach the total of 298 studies as stated. Please verify the search parameters and clarify how this number was obtained.
  • Why are studies up to the present date not included? This omission is a limitation, as it excludes all studies from 2024, potentially impacting the comprehensiveness of the review.
  • I believe the term “schoolchildren” should be incorporated into the advanced search rather than being applied as an additional limitation.
  • There is a lack of a limitations statement or section in the discussion. Please address this by explicitly outlining the study's constraints.

Minor Issues:

  • Line 145: The formatting of "and the inconsistency index (I2)" should be corrected so that the "2" appears as a superscript (I²).

Results

The results are presented in the recommended manner, although the studies included may not be the most appropriate or sufficient.

Discussion

The authors primarily discuss the results, and their significance based on the statistical data. However, they do not mention the training load parameters or the mechanisms responsible for improvements in physical fitness. It is established that until preadolescence, improvements mainly stem from neurological factors, whereas after adolescence, the maturation of the endocrine system influences physical abilities.

Author Response

Response to Reviewer 1 Comments

1. Summary

3. Point-by-point response to Comments and Suggestions for Authors

Comments 1: The article explores the intriguing topic of the impact of strength training on children and adolescents concerning physical fitness parameters. The title suggests that the article will examine studies related to strength training in children at various developmental stages, regardless of the training context. From the introduction, it appears that the study will assess data from research conducted in a school environment. However, the methodology indicates that studies involving children aged 6-16 years will be analyzed, regardless of whether the research was conducted in a school setting or not.

I believe the aim of the article is unclear. In any case, it would be particularly interesting to analyze studies conducted in the school environment, providing physical education teachers the opportunity to adapt their lesson plans accordingly. Furthermore, the discussion was expected to examine the load parameters and draw conclusions regarding the frequency of training, intensity, volume, and rest intervals. Nevertheless, the discussion focused on whether the results were significant.

Response 1: We appreciate this observation, as it correctly identifies a potential ambiguity between the general objective and the context of application of the study. As a result, we have reformulated the objective in both the abstract and the introduction to clarify that the analysis includes studies conducted in both school-based and extracurricular settings, with the aim of offering practical implications for implementation in educational contexts

Comments 2: Introduction

The introduction focuses on the effects of strength training within the school environment on physical fitness parameters. However, it does not align with the title and methodology of the research.

The authors must determine whether they wish to analyze studies conducted in school settings or examine children in general, including those who are or are not involved in organized sports forms.

Response 2:. We thank the reviewer for this valuable observation, which highlights an important issue regarding the alignment between the introduction, the title, and the methodological scope of the study. We agree that the original introduction placed disproportionate emphasis on the school environment, which may have led to confusion about the population and settings included in the analysis.

To address this, we have revised the introduction to clarify that the study encompasses strength training interventions conducted in both school-based and extracurricular or sports contexts, involving healthy children and adolescents aged 6 to 16 years. This clarification better reflects the methodological approach used in the selection of studies and ensures consistency with the article’s title and objectives. Furthermore, we have maintained a focus on educational applicability, as many of the findings can be transferred to school-based physical education settings.

These changes can be found on page 2, paragraphs 1 and 6, lines 36–38 and 76–78 of the revised manuscript.

Updated text (excerpt from manuscript):
“...the aim of this study was to conduct a systematic review and meta-analysis to examine the effects of supervised strength training programmes on physical fitness variables—specifically VOâ‚‚max, muscular strength, and sprint performance—in healthy children and adolescents aged 6 to 16 years. The analysis included studies conducted both within school environments and in extracurricular or sports-based contexts, with the goal of providing evidence-based insights that can inform educational practice and contribute to the development of effective physical training strategies in youth populations.”

Comments 3: Methodology

Major Issues:

  • When performing the advanced search in Scopus, I am unable to reach the total of 298 studies as stated. Please verify the search parameters and clarify how this number was obtained.
  • Why are studies up to the present date not included? This omission is a limitation, as it excludes all studies from 2024, potentially impacting the comprehensiveness of the review.
  • I believe the term “schoolchildren” should be incorporated into the advanced search rather than being applied as an additional limitation.
  • There is a lack of a limitations statement or section in the discussion. Please address this by explicitly outlining the study's constraints.

Minor Issues:

  • Line 145: The formatting of "and the inconsistency index (I2)" should be corrected so that the "2" appears as a superscript (I²).

Response 3:.

We thank the reviewer for these detailed and constructive observations regarding the methodology. In response, we have carefully revised the relevant sections of the manuscript and provide clarifications below:

Scopus search discrepancy: We acknowledge the confusion regarding the number of studies retrieved from Scopus. The original figure of 298 referred to an earlier iteration of the search prior to applying updated filters across databases. After harmonising search terms and applying inclusion criteria (date range, language, population, intervention type, etc.), the total number of retrieved records across EBSCO, Web of Science, and Scopus was 634, as accurately reflected in the revised manuscript (Section 2.1).

Cut-off at 2023: The last search was conducted in December 2023, prior to initiating the analysis phase in early 2024. We agree that this limits the inclusion of studies published in 2024. This has now been explicitly acknowledged as a limitation in the revised Discussion section

Use of “schoolchildren” in the search strategy: As indicated in Section 2.1 (lines 91–94), the term “schoolchildren” was not used directly in the initial search string to avoid excluding relevant studies conducted in non-school contexts (e.g., clubs or extracurricular settings) involving school-aged participants. Instead, this criterion was applied during the full-text screening phase. This strategy has been clarified in the revised manuscript (Section 2.1).

Statement of limitations: A new paragraph outlining the key limitations of the review has been added to the Discussion section . This includes the temporal cut-off, language restrictions, the broad age range, and the absence of subgroup analysis by biological maturation.

Formatting of I²: The formatting issue has been corrected, and “I2” now appears properly as “I²” in Section 2.4

PROSPERO registration: In response to reviewer feedback and to enhance methodological transparency, the review protocol has now been retrospectively registered in PROSPERO (Registration ID: CRD420251029311). A reference to this registration has been added in the Materials and Methods section

Comments 4: Results

The results are presented in the recommended manner, although the studies included may not be the most appropriate or sufficient

Response 4:.

We thank the reviewer for this comment. All included studies were selected based on predefined eligibility criteria using the PICOS framework, including target age range (6–16 years), supervised strength training, and reporting of pre- and post-intervention data on physical fitness outcomes. In addition, all studies were assessed using the AXIS tool and, when applicable, the PEDro scale, ensuring a rigorous methodological appraisal. We acknowledge that the number and nature of the studies included may limit the generalisability of the findings. For this reason, we have added a dedicated Limitations paragraph in the Discussion section, where we explicitly address the constraints related to sample characteristics, language, publication date, and absence of subgroup analysis

Comments 5: Discussion

The authors primarily discuss the results, and their significance based on the statistical data. However, they do not mention the training load parameters or the mechanisms responsible for improvements in physical fitness. It is established that until preadolescence, improvements mainly stem from neurological factors, whereas after adolescence, the maturation of the endocrine system influences physical abilities.

Response 5:.

We thank the reviewer for this insightful observation. In response, we have expanded the Discussion section to address two key aspects. First, we now include a synthesis of the most commonly reported training load parameters (frequency, intensity, volume, and rest) among the included studies, providing practical context for the observed effects. Second, we have added a physiological interpretation of the training responses based on age and biological maturation. Specifically, we discuss how strength gains in prepubescent children are predominantly driven by neuromuscular adaptations (e.g., increased motor unit recruitment and improved coordination), whereas in adolescents, endocrine factors such as increased testosterone and growth hormone contribute to muscular hypertrophy and performance improvements.

These additions can be found in the revised Discussion section (page 13, lines 303–316), and are supported by relevant literature [5, 8, 44, 45].

Reviewer 2 Report

Comments and Suggestions for Authors

Introduction

The introduction provides a general overview on strength training for childhood and adolescent development. The text is clear and well structured, however, I believe the introduction is overly broad and at times lacks focus about the specific objective of the study. In particular, the inclusion of topics such as emotional and social development, academic performance, and mental health, although relevant for youth health, are not directly addressed in the methodology or the results of the review, which focuses exclusively on physiological and performance variables (VOâ‚‚max, muscular strength, sprint performance). This could create a certain misalignment between the introduction and the actual focus of the study, because too much emphasis is placed on topics that are not analysed. So, I suggest reducing the section dedicated to psychological and social benefits highlighting that physical variables are the focus of the meta-analysis.

Methods

Both English and original Spanish-language articles were included in the review, which could introduce potential bias or inconsistencies in interpretation and data extraction.

The databases mentioned in the abstract (EBSCO, Web of Science, and PubMed) do not match those reported in the methods section (EBSCO, WoS, SCOPUS), creating uncertainty about the search strategy adopted.

The inclusion period was limited to articles published between 2013 and 2023, potentially excluding relevant studies published in 2024. I suggest to update the literature review.

Moreover, there is no mention of the involvement of a third reviewer in cases of disagreement during the article selection, which raises concerns about how discrepancies were handled; however, in the section “quality assessment data extraction”, this presence  is reported. The authors should clarify whether a third reviewer was involved throughout the entire review process.

The age range considered (6–16 years), is too large, considering that adolescents between 12 and 16 years old—particularly females—are undergoing pubertal development. No subgroup analyses were conducted, despite the significant physiological and maturational differences between childhood and adolescence.

Finally, the review does not report whether the protocol was registered on PROSPERO. The authors should indicate clearly whether such registration was performed.

Quality assessment data extraction

The PEDro scale is specifically designed for randomized controlled trials (RCTs), yet the authors apply it to all studies in the methodology section. Are all the included studies RCTs? The authors should clarify this aspect.

Line 118: It contains typographical errors. Please check

Table 2: I recommend placing Table 2 immediately after the 'Data Extraction' section.  Also, I recommend revising the structure of the Table 2:

  • Currently, the study design, sample characteristics, and age data are presented together in a single block of text. I suggest separating these elements into two distinct columns: the first should include the study design, and the second should report participants' age and sample size. In addition, I suggest standardizing the way age-related data are reported: both mean age and standard deviation should be included across all studies. If such data are not reported in some of the original studies, a note indicating "not reported" or "NR" should be added.
  • Both the legend and the text in the table includes a mix of Spanish and English terms. I strongly recommend that the entire table be translated fully into English.
  • In several entries, the separation between different studies could be improved. I suggest ensuring consistent and clear spacing between author names to avoid confusion.
  • The list of acronyms at the bottom of the table appears disorganized. I recommend that the acronyms be reorganized in alphabetical order.
  • Please revise the use of the acronym “E” in the table legend. Currently, it is used to indicate both “E = experimental group” and “E = age of the sample”, which creates confusion. These two meanings should not share the same acronym; consider using distinct abbreviations to avoid ambiguity.
  • There are inconsistencies between the acronyms used in the table and those listed in the legend. For example, 'GCON' appears in the table but is not explained in the legend.
  • The acronym “D = duration” appears twice in the table legend. Please review and correct this duplication.

Results:

The results section The results section is well structured and clearly divided by outcome. However, I recommend  few improvements. Rather than repeating the explanation of forest plot elements in each results subsection, I suggest moving that description to the figure legends. This would reduce redundancy in the text and improve readability.

Figures: 

In Figure 2, 3, 4 and 5 some studies appear to be included twice (Alves 2016 and Marta 2013). If these represent distinct data sets or subgroups from the same publication, this should be explicitly clarified in the figure legend or the methodology section. Otherwise, the authors should ensure that duplicate data are not included, as could represent a bias in the meta-analysis results.

Discussion:

I suggest rephrasing the discussion section, as it currently appears too descriptive. In addition to reporting statistical significance, it would be helpful to discuss the clinical or practical relevance of the observed effects—for example, whether the improvements in VOâ‚‚Max or strength have an impact on the physical development of the participants.

Moreover, considering the wide age range of the studied population (6–16 years), I believe it is essential to discuss the influence of age and biological maturation on the measured outcomes. The physiological differences between children and adolescents can substantially affect the effectiveness of strength training. The absence of subgroup analyses makes it even more important to address this variable within the discussion.

Conclusion:

In the conclusions, the authors recommend the inclusion of strength training programs with a minimum duration of 8 weeks. However, this recommendation does not appear to be fully supported by the analysis presented. The duration of the interventions included in the study varies considerably, with some programs lasting less and others more than 8 weeks. I suggest that the authors clarify the basis for this recommendation.

Author Response

Comments 1: Introduction

The introduction provides a general overview on strength training for childhood and adolescent development. The text is clear and well structured, however, I believe the introduction is overly broad and at times lacks focus about the specific objective of the study. In particular, the inclusion of topics such as emotional and social development, academic performance, and mental health, although relevant for youth health, are not directly addressed in the methodology or the results of the review, which focuses exclusively on physiological and performance variables (VOâ‚‚max, muscular strength, sprint performance). This could create a certain misalignment between the introduction and the actual focus of the study, because too much emphasis is placed on topics that are not analysed. So, I suggest reducing the section dedicated to psychological and social benefits highlighting that physical variables are the focus of the meta-analysis

Response 1: We agree with this comment.

We appreciate the reviewer’s thoughtful comment regarding the alignment between the introduction and the specific focus of the study. In response, we have revised the introduction to reduce the emphasis on emotional, social, and academic benefits of strength training, and to clarify that these dimensions are not part of the current analysis. While a brief reference to broader psychosocial outcomes has been retained to contextualise the importance of physical activity in youth development, we now explicitly state that the aim of this systematic review and meta-analysis is to examine the effects of supervised strength training on physical fitness variables only—namely VOâ‚‚max, muscular strength, and sprint performance.

These changes can be found in the Introduction.

Comments 2: Methods

Both English and original Spanish-language articles were included in the review, which could introduce potential bias or inconsistencies in interpretation and data extraction.

The databases mentioned in the abstract (EBSCO, Web of Science, and PubMed) do not match those reported in the methods section (EBSCO, WoS, SCOPUS), creating uncertainty about the search strategy adopted.

The inclusion period was limited to articles published between 2013 and 2023, potentially excluding relevant studies published in 2024. I suggest to update the literature review.

Moreover, there is no mention of the involvement of a third reviewer in cases of disagreement during the article selection, which raises concerns about how discrepancies were handled; however, in the section “quality assessment data extraction”, this presence  is reported. The authors should clarify whether a third reviewer was involved throughout the entire review process.

The age range considered (6–16 years), is too large, considering that adolescents between 12 and 16 years old—particularly females—are undergoing pubertal development. No subgroup analyses were conducted, despite the significant physiological and maturational differences between childhood and adolescence.

Finally, the review does not report whether the protocol was registered on PROSPERO. The authors should indicate clearly whether such registration was performed

Comments 1: [Paste the full reviewer comment here.]

Response 2: Thank you for pointing this out

We acknowledge the reviewer’s concern regarding the inclusion of both English and Spanish-language articles. To minimise potential bias, all articles were assessed using standardised procedures and validated tools (AXIS and PEDro), and data extraction was carried out independently by two reviewers. This approach ensured consistency in interpretation regardless of the language of publication

The correct databases used in the review were EBSCO, Web of Science (WoS), and Scopus. PubMed was mentioned in the original abstract in error and has now been removed to ensure consistency with the methodology section.

We agree that limiting the search to December 2023 may have excluded recent studies. This time frame was necessary to allow for full screening, analysis, and peer review. We have acknowledged this in the Limitations section of the Discussion

A third reviewer was indeed involved in resolving discrepancies throughout the entire review process, including study selection and quality assessment. We have updated the Methods section to reflect this.

We acknowledge the reviewer’s comment regarding the wide age range. Due to the scope and reporting limitations of the included studies, subgroup analyses based on age or biological maturation were not feasible. However, this has been addressed as a limitation in the Discussion

In response to the reviewer’s suggestion, we have now registered the review protocol in PROSPERO to enhance transparency and methodological rigour. The registration number has been added in the Methods section

Comments 3: Quality assessment data extraction

The PEDro scale is specifically designed for randomized controlled trials (RCTs), yet the authors apply it to all studies in the methodology section. Are all the included studies RCTs? The authors should clarify this aspect

Response 3: Agree. We have, accordingly, modified…..to emphasize this point.

We thank the reviewer for this important and accurate observation. We acknowledge that the PEDro scale is specifically designed for randomised controlled trials (RCTs) and should not be applied to studies with other designs. In our original version, this distinction was not clearly stated.

To address this, we have clarified in the revised manuscript that the PEDro scale was used only for studies identified as RCTs, and not for non-randomised or quasi-experimental designs. For all studies—regardless of design—the AXIS tool was used as the primary instrument to assess methodological quality and risk of bias.

Additionally, we now specify that the mean PEDro score of 7.81 refers exclusively to the RCTs included in the review. This clarification strengthens the methodological transparency of our work. The revised text can be found in the Methods section.

Comments 4:

Line 118: It contains typographical errors. Please check

Table 2: I recommend placing Table 2 immediately after the 'Data Extraction' section. 

Also, I recommend revising the structure of the Table 2:

  • Currently, the study design, sample characteristics, and age data are presented together in a single block of text. I suggest separating these elements into two distinct columns: the first should include the study design, and the second should report participants' age and sample size. In addition, I suggest standardizing the way age-related data are reported: both mean age and standard deviation should be included across all studies. If such data are not reported in some of the original studies, a note indicating "not reported" or "NR" should be added.
  • Both the legend and the text in the table includes a mix of Spanish and English terms. I strongly recommend that the entire table be translated fully into English.
  • In several entries, the separation between different studies could be improved. I suggest ensuring consistent and clear spacing between author names to avoid confusion.
  • The list of acronyms at the bottom of the table appears disorganized. I recommend that the acronyms be reorganized in alphabetical order.
  • Please revise the use of the acronym “E” in the table legend. Currently, it is used to indicate both “E = experimental group” and “E = age of the sample”, which creates confusion. These two meanings should not share the same acronym; consider using distinct abbreviations to avoid ambiguity.
  • There are inconsistencies between the acronyms used in the table and those listed in the legend. For example, 'GCON' appears in the table but is not explained in the legend.
  • The acronym “D = duration” appears twice in the table legend. Please review and correct this duplication.

Response 4:

Table 2 and Typographical Corrections

We thank the reviewer for the detailed and constructive feedback:

• The typographical error on line 118 has been reviewed and corrected.

• Table 2 has been moved to immediately follow the “Data Extraction” subsection for greater consistency and clarity.

• To improve readability, the information related to study design, sample size, and age has been reformatted within a single cell using a block structure with explicit labels (e.g., “Design”, “Sample”, “Age”), avoiding the need for additional columns while ensuring clarity.

• Age data have been standardised across studies (mean ± SD), and “NR” has been indicated where not reported.

• The entire table, including all column headings, content, and the legend, has been translated fully into English.

• Visual separation between studies has been improved for consistency.

• The legend has been reorganised alphabetically, redundant entries (such as the duplication of “D = duration”) have been removed, and all acronyms used in the table are now clearly defined.

• To avoid ambiguity, the acronym “E” is now used exclusively for “EG = Experimental group”, and “Age” is used explicitly to describe participant age. The acronyms “GCON”, “ECoN”, and “RYF” have also been added to the legend for completeness.

Comments 5: Results:

The results section The results section is well structured and clearly divided by outcome. However, I recommend  few improvements. Rather than repeating the explanation of forest plot elements in each results subsection, I suggest moving that description to the figure legends. This would reduce redundancy in the text and improve readability.

Figures:

In Figure 2, 3, 4 and 5 some studies appear to be included twice (Alves 2016 and Marta 2013). If these represent distinct data sets or subgroups from the same publication, this should be explicitly clarified in the figure legend or the methodology section. Otherwise, the authors should ensure that duplicate data are not included, as could represent a bias in the meta-analysis results.

Response 5:

Results and Figures

We appreciate the reviewer’s thoughtful observations regarding the presentation of forest plots and the potential duplication of studies.

Regarding the explanation of forest plot elements, we agree that repetition may reduce the clarity of the text. Accordingly, we have removed these descriptions from the Results section and relocated them to the legends of Figures 2–5, where they now appear in a concise and standardised format.

Concerning the repeated appearance of Alves et al. (2016) and Marta et al. (2013) in the forest plots, we confirm that each entry represents a distinct intervention group reported in the same publication. In both cases, the authors present independent experimental conditions with different training interventions applied to different groups, and separate outcome data are reported for each group. Therefore, their inclusion as individual entries in the meta-analysis is methodologically justified and does not constitute data duplication.

While we understand the reviewer’s concern, we believe that disaggregating these results respects the statistical independence of the groups and provides a more detailed picture of the effects of different types of strength training interventions. For transparency, we have added a brief clarification in the Methods section and in the figure legends, indicating that multiple entries from a single study correspond to distinct intervention arms with non-overlapping participants

Comments 6: Discussion:

I suggest rephrasing the discussion section, as it currently appears too descriptive. In addition to reporting statistical significance, it would be helpful to discuss the clinical or practical relevance of the observed effects—for example, whether the improvements in VOâ‚‚Max or strength have an impact on the physical development of the participants.

Moreover, considering the wide age range of the studied population (6–16 years), I believe it is essential to discuss the influence of age and biological maturation on the measured outcomes. The physiological differences between children and adolescents can substantially affect the effectiveness of strength training. The absence of subgroup analyses makes it even more important to address this variable within the discussion.

Conclusion:

In the conclusions, the authors recommend the inclusion of strength training programs with a minimum duration of 8 weeks. However, this recommendation does not appear to be fully supported by the analysis presented. The duration of the interventions included in the study varies considerably, with some programs lasting less and others more than 8 weeks. I suggest that the authors clarify the basis for this recommendation.In Figure 2, 3, 4 and 5 some studies appear to be included twice (Alves 2016 and Marta 2013). If these represent distinct data sets or subgroups from the same publication, this should be explicitly clarified in the figure legend or the methodology section. Otherwise, the authors should ensure that duplicate data are not included, as could represent a bias in the meta-analysis results.

Response 6:

– Discussion

We thank the reviewer for this valuable comment. In response, we have restructured and expanded the Discussion section to address the clinical and practical relevance of the findings. We now highlight how the observed improvements in VOâ‚‚max and muscular strength can contribute to better cardiovascular health, postural stability, injury prevention, and long-term physical literacy in children and adolescents.

We also added a section discussing the role of age and biological maturation in training adaptations. Specifically, we distinguish between neuromuscular improvements in prepubescent children and endocrine-driven hypertrophic responses during and after puberty. Although subgroup analyses were not feasible due to data limitations, we acknowledge this as a limitation and encourage future studies to stratify participants by biological maturity. These changes can be found in the Discussion section

– Conclusion

We agree that our original recommendation of a minimum 8-week programme required clarification. In the revised conclusion, we now state that most effective programmes ranged from 6 to 12 weeks, and that measurable improvements were typically observed in interventions lasting at least 6 to 8 weeks. This conclusion is supported both by the analysis of included studies and by previous literature on training adaptations in youth. The updated statement can be found in the Conclusion section

– Figures

As explained above , Alves et al. (2016) and Marta et al. (2013) reported data from multiple independent intervention groups. These were analysed separately and do not represent duplicated data. This has been clarified in the Methods section and in the figure legends of Figures 2, 4, and 5

Round 2

Reviewer 1 Report

Comments and Suggestions for Authors

The authors of the article made a considerable effort to address the reviewers' comments. However, several questions remain unanswered.

The title is somewhat general relative to the content of the article.

Over time, numerous studies have examined the effect of strength training on physical fitness parameters, illustrating its significance and resulting in guidelines for developing effective strength programs.

The study should be framed in the context of implementing strength programs in the school environment, which would indeed be innovative.

The introduction has improved significantly; however, if the study gains direction, it should be enriched.

The methodology has been upgraded. However, the fact that the research is limited to the year 2023 poses a significant limitation. Surely, the authors could repeat the search to enrich the results.

In the results, it is suggested that the presentation of the studies in the tables be organized by physical performance ability (endurance, strength, speed). Additionally, a study (Amero et al., 2017) is noted that does not specify the age of the children, which provides a reason for its exclusion from the review. Furthermore, the results mention the term "sem" in relation to the duration of the study, which I do not understand.

The studies whose duration is mentioned in weeks are found to last for several weeks. The studies could be directed towards medium-term adaptations.

The discussion could change if the research is oriented towards a context such as the school environment or the duration of the study.

In conclusion, the authors have made an effort to address all the comments; however, additional work is needed for the study to establish its identity and be valuable to readers.

Author Response

1. Summary

3. Point-by-point response to Comments and Suggestions for Authors

Comments 1:

The title is somewhat general relative to the content of the article

Response 1: We appreciate the reviewer’s observation regarding the general nature of the original title. In response, we have refined the title to more precisely reflect the focus of the study. The revised title now highlights both the type of intervention ("supervised strength training") and the outcomes assessed ("physical fitness"), while maintaining clarity and coherence with the study’s scope and methodology.

The updated title is:

“Effects of Supervised Strength Training on Physical Fitness in Children and Adolescents: A Systematic Review and Meta-Analysis”

We believe this revision better communicates the content and value of the review to readers.

Comments 2:

Over time, numerous studies have examined the effect of strength training on physical fitness parameters, illustrating its significance and resulting in guidelines for developing effective strength programs.

Response 2:. We thank the reviewer for this valuable observation regarding the need to better contextualise our study within the existing body of literature. In response, we have revised the final paragraph of the Introduction to explicitly acknowledge the existence of prior position statements and guidelines on youth strength training [5,8].

Specifically, we now emphasise that while these previous works have provided general recommendations on programme design and safety, the present review offers additional value by delivering an updated quantitative synthesis of the effects of supervised strength training on key physical fitness outcomes (VOâ‚‚max, muscular strength, and sprint) over the last decade.

This addition strengthens the theoretical foundation of the manuscript and clarifies how our study builds upon established frameworks to provide practical, evidence-based insights for both educational and training contexts. The updated content appears in the final paragraph of the Introduction

Comments 3:

The study should be framed in the context of implementing strength programs in the school environment, which would indeed be innovative

Response 3:.

We appreciate the reviewer’s suggestion to frame the study specifically within the context of implementing strength programmes in the school environment. We fully agree that school-based training is a relevant and promising area, and we have carefully considered this perspective. However, we chose to maintain a broader scope that includes both school-based and extracurricular interventions.

This decision is based on the understanding that, in youth populations (ages 6–16), the structure, intensity, and objectives of supervised strength training programmes in school and sports settings are often comparable. Sessions in both contexts typically last 30–45 minutes, focus on neuromuscular development and physical literacy, and use similar formats (e.g., circuit training, bodyweight exercises, resistance bands).

By including both environments, our study provides a more comprehensive synthesis of current practice and outcomes in youth strength training. Importantly, many of the protocols analysed are directly transferable to physical education settings. We have reinforced this practical applicability in the Discussion and Conclusion sections, without limiting the generalisability or value of the review.

We hope this clarification explains the rationale behind our framing and underlines our commitment to maintaining both methodological rigour and real-world relevance.

Comments 4:

The introduction has improved significantly; however, if the study gains direction, it should be enriched.

Response 4:.

We sincerely thank the reviewer for acknowledging the improvements made to the Introduction. In response to previous feedback, we have already expanded and refined this section to better define the study’s objective and contextualise its contribution relative to existing guidelines on youth strength training.

While we appreciate the suggestion to further enrich the Introduction should the study gain a more specific direction, we believe that the current version achieves a balance between clarity, academic rigour, and the constraints of the journal’s editorial format.

In particular, we have aimed to maintain conciseness while ensuring that the rationale, scope, and relevance of the study are clearly articulated for both academic and practitioner audiences. We hope this version meets the expectations of the editorial team and readers, while respecting the manuscript length limitations.

The methodology has been upgraded. However, the fact that the research is limited to the year 2023 poses a significant limitation. Surely, the authors could repeat the search to enrich the results.

We thank the reviewer for this thoughtful observation. The literature search was finalised in December 2023, covering a ten-year period (2013–2023) to ensure a robust and up-to-date synthesis of the most recent evidence on supervised strength training in youth.

We acknowledge that limiting the review to studies published up to the end of 2023 may exclude some recently published research; this has been noted in the Limitations section (page 14, lines 327–330). However, we consider that the chosen timeframe remains scientifically valid, as it captures the most methodologically relevant and applicable evidence available at the time of analysis.

Rather than compromise consistency and comparability across included studies, we opted to preserve a clearly defined and relevant timeframe, in line with standard systematic review practices. Future updates of this review may incorporate studies published beyond 2023 to further enrich the field.

Comments 5:

In the results, it is suggested that the presentation of the studies in the tables be organized by physical performance ability (endurance, strength, speed).

Response 5:.

We thank the reviewer for this thoughtful suggestion. Rather than reorganising the table or adding a new column, we have opted to maintain the current structure, as the key outcome domains (e.g., VOâ‚‚max, muscular strength, sprint performance) are already clearly indicated in the “Variables” and “Instruments” columns for each study.

We believe this approach ensures both clarity and consistency, while preserving the layout and readability of the table in accordance with the journal’s formatting guidelines. The alphabetical order also facilitates quick access to each study, and readers can easily identify the performance domains assessed in each case

Comments 6:

Additionally, a study (Amero et al., 2017) is noted that does not specify the age of the children, which provides a reason for its exclusion from the review

Response 6:

We appreciate the reviewer’s attention to detail. Upon rechecking, we confirm that the study by Amaro et al. (2017) does report the age of the participants in both the text and Table 1. The mean age for all groups was 12.7 ± 0.8 years, placing the participants well within the defined inclusion range (6–16 years). Therefore, we believe this study meets the eligibility criteria and remains valid for inclusion in the review.

We will ensure that this information is clearly indicated in Table 2, and the “Age” field has been updated accordingly to avoid confusion.

Comments 7:

Furthermore, the results mention the term "sem" in relation to the duration of the study, which I do not understand.

Response 7:

We thank the reviewer for this helpful observation. In response, we have thoroughly reviewed and revised the manuscript and tables to replace all instances of the abbreviation “sem” with the full English term “weeks”. This change has been applied consistently throughout the text, including in Table 2 and the Results section, to ensure clarity and terminological precision

Comments 8:

The studies whose duration is mentioned in weeks are found to last for several weeks. The studies could be directed towards medium-term adaptations.

Response 8:

We thank the reviewer for this insightful comment. Indeed, most of the included studies implemented interventions lasting between 6 and 12 weeks, which we agree can be considered as medium-term programmes in the context of youth training and physical education.

In response, we have incorporated this perspective into the Discussion section , emphasising that the physical fitness improvements observed are consistent with expected adaptations in mid-term strength training interventions for children and adolescents.

We believe this clarification enhances the practical interpretation of our findings and supports the relevance of such programmes in both school-based and extracurricular settings

Comments 9:

The discussion could change if the research is oriented towards a context such as the school environment or the duration of the study.

Response 9:

We appreciate the reviewer’s suggestion to further explore the implications of the study within specific contexts such as school environments or programme duration. While the interventions included in this review were conducted in both school-based and extracurricular settings, we have maintained a broader perspective to ensure generalisability and reflect the real-world overlap between these contexts in terms of structure, content, and supervision.

Nevertheless, we have reinforced the practical implications for educational settings in the Discussion and Conclusion sections, especially regarding the feasibility of implementing supervised strength training sessions of moderate duration (6–12 weeks) within school curricula. We have also included additional commentary on these medium-term adaptations, in response to previous feedback.

We believe this framing balances methodological rigour with practical applicability, without narrowing the scope of the review.

Comments 10:

In conclusion, the authors have made an effort to address all the comments; however, additional work is needed for the study to establish its identity and be valuable to readers.

Response 10:

We appreciate the reviewer’s overall assessment and acknowledgement of the improvements made throughout the manuscript. We respectfully believe that the study has now achieved a coherent identity, as a systematic review and meta-analysis focused on supervised strength training in youth, with specific attention to VOâ‚‚max, muscular strength, and sprint performance.

The study integrates evidence from both school-based and extracurricular interventions to provide a comprehensive synthesis of mid-term physical fitness adaptations in children and adolescents. We have clarified its methodological scope, reinforced its practical relevance—particularly in educational settings—and contextualised it within existing guidelines and recent literature.

We hope that these clarifications help strengthen the perceived value and direction of the work for readers and practitioners alike

Round 3

Reviewer 1 Report

Comments and Suggestions for Authors

I confirm that the authors have made changes and upgraded the manuscript.

However, some points remain unclear, and I have a different opinion. For example, how long can strength training last in different contexts, such as School or Sport?

I also believe that referring to the decade up to 2023 and not enriching the work with studies from 2024, as a limitation of the study, is a serious methodological issue that the Editor should decide on. I believe I highlighted the issue three times during the review process, and the authors have a specific opinion.

Author Response

3. Point-by-point response to Comments and Suggestions for Authors

Comments 1:

For example, how long can strength training last in different contexts, such as School or Sport?

Response 1: We thank the reviewer for raising this important point. We acknowledge that the contextual differences between school-based and extracurricular training environments can influence how strength training programmes are designed and delivered. However, based on the analysis of the included studies and the supporting scientific literature, we observed that the duration, frequency, and pedagogical structure of training sessions were highly consistent across both settings.

To address this issue explicitly, we have included the following statement at the end of the Discussion section:

“Furthermore, based on the analysis of the included studies and in line with foundational literature in youth strength training [8,47,48], we identified substantial similarities between extracurricular and school-based training programmes in children and adolescents aged 6 to 16. These similarities include session duration and frequency, the use of playful or pedagogically guided activities, and the emphasis on supervised, age-appropriate training. This conceptual and methodological convergence supports the decision to consider both contexts jointly in this review and reinforces the transferability of the results to educational settings.”

We believe this addition helps clarify the rationale behind treating both environments as methodologically comparable within the scope of our systematic review..

Comments 2:

I also believe that referring to the decade up to 2023 and not enriching the work with studies from 2024, as a limitation of the study, is a serious methodological issue that the Editor should decide on. I believe I highlighted the issue three times during the review process, and the authors have a specific opinion.

Response 2:. We sincerely thank the reviewer for their continued attention to the temporal scope of our systematic review. We acknowledge the importance of this issue and have reflected it explicitly in the manuscript as a limitation, recognising that our search strategy did not include studies published in 2024.

The decision to limit the review to the 2013–2023 period was based on ensuring methodological consistency across the stages of selection, analysis, and synthesis. This approach is common in systematic reviews and aligns with the framework registered in PROSPERO (CRD420251029311), where the temporal cut-off was clearly stated.

While we understand and respect the reviewer’s view, we maintain that our approach remains methodologically sound and transparent. We also agree that future updates to this review could incorporate studies published after 2023 to further enrich the evidence base. We defer to the Editor’s judgment regarding the appropriateness of this decision.
